# An Estimate of the Scale and Composition of the Hedgehog (*Erinaceus europeaus*) Rehabilitation Community in Britain and the Channel Islands

**DOI:** 10.3390/ani12223139

**Published:** 2022-11-14

**Authors:** Lucy E. Bearman-Brown, Philip J. Baker

**Affiliations:** 1Department of Animal & Agriculture, Hartpury University, Gloucestershire GL19 3BE, UK; 2School of Biological Sciences, University of Reading, Reading RG6 6AH, UK

**Keywords:** animal welfare, conservation, *Erinaceus europaeus*, European hedgehog, wildlife hospital, wildlife rehabilitation

## Abstract

**Simple Summary:**

Large numbers of animals enter wildlife hospitals/centres each year around the globe, but it is unclear whether the efforts of wildlife rehabilitators have significant impacts on the conservation of the species involved. In this study, we used a questionnaire survey to estimate the number and characteristics of practitioners helping to rehabilitate injured and orphaned hedgehogs (*Erinaceus europaeus*) in Britain, and the number of hedgehogs admitted in one benchmark year (2016). Overall, 304 rehabilitators were identified: 148 supplied data on their structure, and 174 outlined the number of hedgehogs admitted in 2016. Most hospitals (62.6%) were small (admitting <50 hedgehogs each year), but most hedgehogs (82.8%) were admitted to large hospitals (>250 hedgehogs each year). We estimated that this rehabilitation community collectively admitted >40,000 hedgehogs in the benchmark year, of which approximately half could have survived to be released. Assuming that most hedgehogs originated from urban areas, we estimate that >3% of the post-breeding population of hedgehogs entered wildlife hospitals in 2016. In contrast, the urban hedgehog population in Britain is estimated to have declined by approximately 2% per year during 2003–2017. These figures suggest, therefore, that wildlife rehabilitation has potentially been an important factor in the dynamics of hedgehog populations in this country during this period.

**Abstract:**

The conservation benefits of wildlife rehabilitation are equivocal, but could be substantial for formerly common species that are declining rapidly but are still commonly admitted to wildlife centres. We used a questionnaire survey to estimate the number of practitioners rehabilitating West European hedgehogs (*Erinaceus europaeus*) in Britain and the numbers entering hospitals/centres in one benchmark year (2016); practitioners were identified using an internet search and snowball sampling. Overall, 304 rehabilitators were identified: 148 supplied data on their structure, and 174 outlined the number of hedgehogs admitted in 2016. The former comprised 62.6% small (≤50 hedgehogs admitted year^−1^), 16.7% medium-sized (51–250 yr^−1^), and 20.7% large (>250 yr^−1^) hospitals; however, these accounted for 4.8%, 12.4%, and 82.8% of hedgehog admissions, respectively. Small hospitals were less likely to be registered as a charity, have paid staff, have a social media account, to record admissions electronically, or to conduct post-release monitoring. However, they were more likely to operate from their home address and to have been established for ≤5 years. Extrapolations indicate that this rehabilitation community admitted >40,000 hedgehogs in 2016, of which approximately 50% could have been released. These figures suggest that wildlife rehabilitation has potentially been an important factor in the dynamics of hedgehog populations in Britain in the last two decades.

## 1. Introduction

The International Wildlife Rehabilitation Council defines wildlife rehabilitation (WR) as ‘the treatment and temporary care of injured, diseased, and displaced indigenous animals, and the subsequent release of healthy animals to appropriate habitats in the wild’ [1]. Although specific data are lacking, it is reasonable to assume that the practice of rehabilitating wild animals has increased at an international level as a result of the increasing negative impact of humans on natural ecosystems [2,3,4,5,6,7,8,9]. Although widely perceived as helping wildlife, the role of WR as a conservation tool is contested [10]. For example, it can be argued that most animals that enter wildlife hospitals are from common and widespread species [11,12], and that the money spent on their care cannot then be spent on wider conservation actions such as habitat preservation [13]. Similarly, unless released individuals have survival and/or reproduction rates comparable to those which have not required treatment, the cost-effectiveness of WR could be questioned [1,14,15,16]. Consequently, WR has more often been portrayed as an animal welfare issue or for the “benefit of the individual” [14,17,18,19]. 

However, WR can aid conservation in a more general context, such as aiding education [12,14,16,19,20], disease surveillance [21,22,23,24], monitoring of environmental pollutants [25], and the development of capture, treatment, and release protocols that can subsequently be applied to species of conservation concern [26,27]. In addition, it can help to reduce the welfare and population-level impacts of catastrophic events such as oil spills and wildfires, where large numbers of individuals may be affected in a very short space of time [28,29,30,31], by, e.g., reducing mortality rates and by euthanising individuals considered likely to die of their injuries. Furthermore, although high conservation status is often perceived in the context of rarity, it can also result from rapid declines of species that were formerly abundant and widespread [32,33]; in this context, wildlife rehabilitators may receive large numbers of individuals whose care could potentially contribute to the conservation of that species. The magnitude of this benefit is, therefore, partially dependent on the number of animals rehabilitated and released in relation to population size [14,34].

Quantifying the numbers of different species which are successfully rehabilitated and released can, however, be challenging because of the way that WR is often practised. In some countries, the wildlife rehabilitation community may encompass single individuals, charitable and non-charitable NGOs, and/or government agencies, some of which may focus on wildlife generally whilst others focus on just one or a few species [11,14,20]. Consequently, some rehabilitators may only treat a handful of animals each year, whereas larger organisations may treat thousands. As such, there may be substantive differences between organisations with respect to size, facilities, operating protocols, and experience which could affect patterns of care and, ultimately, release rates. In addition, not all countries require that wildlife rehabilitators are licenced or registered [20,35], such that even identifying the number of rehabilitators operating at any given time is problematic. This issue is currently of interest in Britain in the context of the rehabilitation of West European hedgehogs (*Erinaceus europeaus*; hereafter ‘hedgehog’), a species of increasing conservation concern. 

The hedgehog is a small (<1.5 kg), insectivorous mammal found throughout western Europe [36]. In Britain, it is found in a wide range of human-dominated landscapes, including arable and pastoral farmland as well as urban areas [37,38,39,40,41,42]. Evidence from a range of different monitoring programs suggests that populations may have declined by up to 40% in some habitats in the last few decades [40,42,43,44], with declines to varying degrees also present throughout Europe [39,45,46,47]. Factors likely to be associated with this decline include: habitat loss, fragmentation, and degradation; the application of chemical biocides; an increase in the size of road networks and associated traffic volume; the increased abundance of an intra-guild predator, the Eurasian badger (*Meles meles*); and climate change [26,46,48,49,50,51,52,53,54,55]. Most recently, the British hedgehog population was estimated to number approximately 0.88 million individuals [44,56], down from 1.56m in the mid-1990s [57]; although the veracity of both estimates is equivocal, these, in combination with the trends outlined above [40,42,43,44], triggered the species’ status to be upgraded to Vulnerable in these countries [56]. Despite this substantial decline, hedgehogs are frequently the most common mammal species admitted to wildlife hospitals in Britain [10,11], and are also commonly taken to veterinary surgeons by members of the public for treatment [58].

### Hedgehog Rehabilitation in Britain

Whilst the specific details of wildlife legislation in Britain are complex, in general terms, members of the public are allowed to take any injured (or orphaned) wild animal into captivity for the purposes of treatment or care prior to its subsequent release. Whilst in captivity, the animal must receive appropriate husbandry and be taken to a veterinary surgeon for examination if necessary [35,59]; as part of this care, hedgehogs may be humanely killed (euthanised) in order to prevent suffering and/or where they are unsuitable for release. At the point the animal is deemed fit enough to survive in the wild, it should be released [1]. For those hedgehogs that cannot be released because they are unlikely to survive, many rehabilitators recommend euthanasia. Conversely, others consider retention in captivity an acceptable option, although some oppose this on welfare, and perhaps legal, grounds [1,4,59,60].

The wildlife rehabilitation community in Britain is large, diverse, and, in some respects, disjointed [10]. At one end of the spectrum is the Royal Society for the Protection of Animals (RSPCA), the largest animal welfare organisation in England and Wales, and the Scottish Society for the Protection of Animals (SSPCA), which operates in Scotland. Both organisations investigate and enforce cases associated with animal welfare and animal cruelty, including wild animals, but also rehabilitate injured wild animals; the RSPCA has four wildlife centres based in England, and the SSPCA has one based in Clackmannanshire. Similarly, Tiggywinkles Wildlife Hospital in Oxfordshire, England is considered the largest purpose-built wildlife hospital in Europe. Individually, these three organisations may each admit >1000 hedgehogs each year. 

However, there are a substantial number of smaller organisations and individuals who also rehabilitate hedgehogs in Britain. For example, the British Hedgehog Preservation Society (BHPS) (http://www.britishhedgehogs.org.uk, accessed on 10 November 2022), a charitable organisation that has a specific focus on hedgehog conservation and rehabilitation, maintains a service whereby members of the public can call them to ask for the contact details of their nearest hedgehog carer/rehabilitator. Under UK data protection rules, this information cannot be disseminated to researchers, but it is estimated that this list may contain up to 600 different individuals and organisations (Fay Vass, President of the BHPS, pers. comm.; [36]). These are often individuals working from their own private residence, with hedgehogs housed within their home itself or in a shed or purpose-built unit in their garden. Given the widespread interest in the plight of hedgehogs in Britain, and the availability of training courses associated with treating and rehabilitating hedgehogs (e.g., http://www.valewildlife.org.uk/courses/, accessed on 10 November 2022), the numbers of these individuals/smaller organisations is likely to have grown in recent years.

Despite earlier recommendations about the scientific merits of collecting and collating data about animals entering wildlife hospitals in Britain [4,61], relatively few data exist on numbers admitted annually. For example, [11] reported an estimated 30,000–40,000 casualties per annum, the most common species being red fox (*Vulpes vulpes*), Eurasian badger (*Meles meles*), hedgehog and blackbird (*Turdus merula*). More recently, [4] reported 71,000 animals were admitted to the RSPCA’s four wildlife centres and 23 other wildlife hospitals in 2011. Given this paucity of information, and the potentially increasing importance of rehabilitation as hedgehog numbers continue to decline, we used a structured internet search and questionnaire survey to estimate (1) the number of individuals and organisations rehabilitating hedgehogs in Britain, and (2) the number of hedgehogs admitted to wildlife hospitals in a single year (2016). The questionnaire also (3) requested information relating to the structure and practices of these individuals and organisations to identify how the British hedgehog rehabilitation community is structured at the current time. In addition, we (4) highlight the challenges associated with deriving estimates of the number of rehabilitators practising and the numbers of hedgehogs admitted in the context of similar future studies. 

## 2. Materials and Methods

For brevity, we use the term ‘rehabilitator’ to refer to any individual or organisation that treats and releases hedgehogs; in Britain, this is also synonymous with the term “carer.” The terms ‘hospital’ or ‘centre’ refer to any building or structure from which a rehabilitator operates (including private households, buildings on private premises such as a garden shed and/or large purpose-build facilities). The terms ‘hospital’ or ‘centre’ do not necessarily reflect the size of an establishment nor the range of medical facilities present.

A systematic internet search was undertaken from September 2016–January 2017 using the online search engines Google and Bing, websites (e.g., www.britishhedgehogs.org.uk, accessed on 28 January 2017), online databases (e.g., www.helpwildlife.co.uk/directory, accessed on 28 January 2017), and the social media platforms Facebook and Twitter to create a database of hedgehog rehabilitators. Search terms included: “wildlife hospital,” “wildlife rescue,” “hedgehog hospital,” “hogspital” and “hedgehog rescue.” Where available, the name and location of each rehabilitator was provisionally recorded if they had an online presence in any form (e.g., social media profile, if they had been mentioned in local or national media, if they had a fundraising campaign). However, because online information may be erroneous (e.g., they had ceased practising), rehabilitators were only classified as active if their online information indicated that they were still operating in 2016; where this information was not immediately evident, the rehabilitator was contacted directly by email or via social media to confirm.

Following this online search, snowball sampling (chain-referral sampling) was used to help identify additional rehabilitators. This was done by searching the social media associates of each provisionally identified rehabilitator, and by also asking them to forward/advertise our survey questionnaire to their contacts. This approach helped to identify rehabilitators that had no obvious online social media presence and individuals who had only recently started practising.

### 2.1. Questionnaire Survey

A self-administered questionnaire was distributed via SmartSurvey^®^ from January 2017–December 2017. The questionnaire was publicised through social media using web pages associated with wildlife and hedgehog rehabilitation, and through contacting the rehabilitators identified above directly. The questionnaire was further publicised via newsletters published by the BHPS, British Wildlife Rehabilitation Council, and the People’s Trust for Endangered Species. The BHPS promoted the questionnaire to all ~600 carers in their database. All centres were contacted at least once via email or a social media message, depending on their preferred mode of communication advertised on their website or social media. Information requested included: their name; whether they were a registered charity or not; how many paid personnel they employed; how many unpaid staff (volunteers) worked at their hospital; whether they had a full-time veterinary surgeon on staff or worked in conjunction with an external veterinary practice; the year they had started rehabilitating hedgehogs; whether their hospital was run from their personal residence or from a purpose-built rehabilitation centre; whether they had a personal and/or business social media account for advertising their hospital to the general public; and whether they used paper records or a computer to record information about the animals they have cared for.

Respondents were then asked to indicate how many hedgehogs they had received each year for the 5-year period from 2012–2016, inclusive; as 2016 was the most recent year for which respondents would have had complete information, this was taken as the benchmark year for estimating the number of animals admitted. Finally, respondents were asked to indicate whether they undertook post-release monitoring of any sort and, if so, what methods they used (radio-tags, GPS-tags, injected microchips (PIT tags), ear tags, marking spines with numbered tags (e.g., [62] or nail varnish). Such monitoring is typically passive, as it is focused on identifying animals that are readmitted to hospitals after their release, individuals that have been killed on roads near to where they were released, and/or the use of motion-activated video cameras at release sites.

### 2.2. Estimating the Number of Hedgehog Admissions

In order to estimate the total number of hedgehogs admitted in 2016 by all active hospitals identified, we first categorised those hospitals for which we had data on the number of admissions into three size classes: small, medium, and large. These divisions were estimated retrospectively based upon the frequency distribution of the numbers of hedgehogs admitted (see Results). These were assigned both to reflect the pattern of admissions but also to ensure that a sufficient number of hospitals (both from the original searches and from snowball sampling) was in each division to enable statistical analysis. Differences in the relative numbers of hospitals in each size class identified in the online searches versus the snowball sampling were compared using a chi-squared test. Differences in the median number of hedgehogs admitted in 2016 within each size class in the online searches versus snowball sampling were compared using a series of Mann–Whitney tests.

Two models were used to estimate the numbers of hedgehogs admitted by those hospitals for which there were no data available. In Model 1, the data from both the online searches and snowball sampling were merged and treated as a homogenous sample. These combined data were used to estimate the proportion of small (PS), medium (PM), and large (PL) hospitals in the overall community, as well as the median number of hedgehogs admitted by each hospital in each size class (NS, NM, and NL, respectively). These values were then used to estimate the number of hedgehogs (TX) admitted to those hospitals with missing data (H): TX = H × PM × NS + H × PM × NM + H × PL × NL. These were then added to the number known to have been admitted to those hospitals for which data were available (TY) in order to estimate the total number of hedgehogs admitted in 2016 (T): T = TX + TY.

Model 2 followed a similar approach, except that the data from the online searches and snowball sampling were treated separately, as there was evidence that the composition of each sample varied with respect to the proportion of hospitals in each size class and the median number of hedgehogs admitted within each size class (see Results). 

### 2.3. Structure of the Rehabilitation Community

Differences in the characteristics of hospital size classes were quantified using data from those rehabilitators where we had both an estimate of their size and who had completed the questionnaire survey; rehabilitators who had completed the questionnaire but who had not indicated the numbers of hedgehogs admitted in 2016 were excluded. Similarly, rehabilitators who had failed to answer a specific question were excluded from the analysis relating to that variable.

A series of chi-squared tests were used to compare differences between the three hospital classes with respect to: (i) whether they were a registered charity or not; (ii) the type of veterinary care they had (five categories: none, work with an external veterinary practice, work with external wildlife hospital, they themselves are a veterinary nurse or veterinary surgeon, onsite veterinary surgeon); (iii) how long they had been established (data merged into two categories: ≤5 years and >5 years); (iv) where hedgehogs were housed during rehabilitation (three categories: building in private grounds, e.g., garden shed; in their private residence; a purpose-built facility); (v) the type of social media account(s) that they had (three categories: none; only one or more personal social media accounts; one or more business accounts, with or without personal accounts as well); (vi) how they kept records of the hedgehogs admitted (two categories: fully or partly on paper; fully or partly on computer. NB: the option “partly paper” indicates that the majority of records were recorded on paper with a minority on computer, whereas “partly computer” indicates the reverse); and (vii) whether they did or did not undertake any form of post-release monitoring.

In addition, we quantified the number of paid and unpaid (volunteers) staff working in each hospital size class. The latter was divided into three categories (1 volunteer, 2 volunteers, ≥3 volunteers) and analysed using a chi-squared test; where hospitals were run by just one unpaid person, this would be the person in charge of that hospital who was running it on a voluntary basis. It was not possible to statistically compare the number of paid staff, as very few hospitals employed any paid personnel. Therefore, we have simply summarised these data by indicating: the percentage of hospitals within each size class that employed one or more paid staff; and the mean number of paid staff in those hospitals where they were present.

Temporal trends in the annual number of admissions were investigated in order to identify whether 2016 was potentially an atypical year using two approaches. First, data from the current questionnaire were used to compare the median number of hedgehogs admitted in each hospital size class for the five-year period from 2012–2016, inclusive. Median (±IQR) numbers of admissions are presented for small, medium, and large hospitals separately, utilising all data available. However, not all hospitals supplied data for all five years; therefore, separate Friedman tests were used to identify between-year differences using those subsets of hospitals within each size class that did supply data for all five years. Second, data on the number of juveniles, adults, and hedgehogs of unknown age admitted to the four RSPCA wildlife centres in England (East Winch, Norfolk; Mallydams Wood, East Sussex; Stapeley Grange, Cheshire; West Hatch, Somerset) were collated for the period from January 2006 to December 2018, inclusive.

All statistical analyses were conducted using MINITAB version 19.1.1 and SPSS version 25. Nonparametric tests were used throughout, as the data were not normally distributed. Data are therefore presented as medians ± inter-quartile ranges unless otherwise specified.

## 3. Results

Overall, 239 rehabilitators were provisionally identified through online searches; 179 were considered active in 2016, 47 were considered not active, and 13 were of unknown status. Of the 179 that were active, information on the number of hedgehogs admitted was available from 59 (33.0%). A further 125 rehabilitators were identified by snowball sampling, all of which were considered active; 115 (92.0%) provided information on admissions. Data on the number of admissions in the benchmark year were available for 174 of 304 hospitals (57.2%: *n* = 59 + 115), but with data missing from a further 130 known establishments (*n* = 120 + 10).

Based on the pattern of admissions in 2016 (Figure 1), small, medium, and large hospitals were defined as those which admitted ≤50 (n = 109: 62.6%), 51–250 (n = 29: 16.7%), and >250 (n = 36: 20.7%) hedgehogs, respectively. Significantly more small hospitals and fewer large hospitals were detected by snowball sampling compared to the original online search (Chi-squared test: X^2^_2_ = 67.18, *p* < 0.001; Figure 1).

### 3.1. Estimating Total Hedgehog Admissions in Benchmark Year

Overall, the 174 rehabilitators for which data were available admitted 25,540 hedgehogs in 2016, with large hospitals collectively dealing with substantially larger numbers (n = 21,145; 82.8%), than medium-sized (n = 3169; 12.4%) or small (n = 1226; 4.8%) organisations. Significantly fewer hedgehogs were admitted to small hospitals depending on whether they had been identified by snowball sampling versus those that had been identified in the original online search (Mann–Whitney test: W = 1124.00, *p* < 0.001; Table 1); no significant differences were evident for medium-sized (W = 256.50, *p* = 0.965) or large (W = 578.50, P = 0.097) hospitals. Extrapolating from these data, the number of hedgehogs admitted by all 304 active hospitals in the combined sample was estimated to range from 40,991 (Model 1) to 59,308 (Model 2) individuals (Table 2).

### 3.2. Structure of the Hedgehog Rehabilitation Community

Overall, 148 of the 304 active rehabilitators contacted (48.7%) supplied information about both their structure and the number of hedgehogs they admitted in 2016. However, not all respondents answered all questions about their composition, so sample sizes varied between analyses.

Hospitals varied significantly with respect to their charitable status, the number of unpaid staff working at the hospital, the length of time they had been established, where hedgehogs were housed during the rehabilitation process, their social media presence, patterns of record-keeping but not patterns of veterinary care, and whether they conducted post-release monitoring (Table 3). In general terms, small and medium-sized hospitals were less likely to be registered as charities, more likely to have been established within the five years prior to 2016 (Figure 2), and more likely to operate out of the rehabilitator’s private residence (Table 3). Furthermore, smaller hospitals were most commonly staffed by just one unpaid person (Figure 3), less likely to have a business social media presence, more likely to rely on paper records, and less likely to carry out post-release monitoring (Table 3). Post-release monitoring by all hospitals was predominantly via the use of spinal tags or nail varnish (n = 65 of 70 hospitals that conducted post-release monitoring). All three categories of hospital relied extensively on support from an external veterinary practice. Paid staff were present in <5% of small and medium-sized hospitals, but >40% of large hospitals (Table 3).

The median number of hedgehogs submitted annually throughout the period from 2012–2016 appeared to increase for small (Figure 4a) and medium-sized (Figure 4b) hospitals, and to a lesser degree for large hospitals (Figure 4c). Considering only those hospitals where there were 5 years’ worth of data (n = 28), the median number of hedgehogs submitted in 2016 was significantly higher than in both 2012 and 2013 (Friedman test: H = 22.94, DF = 4, *p* < 0.001; Figure 4d). There was a significant positive correlation between the number of juveniles and adult hedgehogs admitted to the four RSPCA wildlife centres annually (Pearson correlation coefficient: r = 0.753, *p* < 0.001), with 2016 being associated with a particularly large number of admissions (Figure 5).

## 4. Discussion

This study is, to the best of our knowledge, the first to attempt to estimate the number of practitioners involved in the rehabilitation of hedgehogs in Britain and the number of hedgehogs admitted into their care. At one level, searching for rehabilitators via the internet should be straightforward: as members of the public need to be able to locate and contact individuals or organisations who take in and care for injured or orphaned hedgehogs, it would be expected that practitioners would maintain an active social media presence advertising their services. However, this did not seem to be the case. Overall, we identified 304 active rehabilitators, but only 58.9% were identified in the original online searches; the remainder were only identified by snowball sampling (i.e., relying on provisionally identified practitioners to further advertise our request for information to their personal contacts). This potentially indicates that a large proportion of hedgehog rehabilitators in Britain rely on indirect contact networks (e.g., referrals from other rehabilitators) or “word of mouth” in order to be found by members of the public. This increases the possibility that they will not be identified in studies like this one, such that the numbers presented below should be considered minimum estimates. In addition, this also means that some smaller hospitals may be difficult for members of the public to contact in the event that they discover an injured or orphaned hedgehog.

In terms of the number of hospitals, the hedgehog rehabilitation community in Britain is dominated by small hospitals (62.6%), with many fewer medium-sized (16.7%) and large (20.7%) establishments. This does, in part, reflect the approach we used to group hospitals into different size classes, but it is clear that a very large number of rehabilitators deal with relatively small numbers of admissions annually (Figure 1). This pattern is further reflected by a wide range of associated characteristics. For example, small hospitals were less likely to be registered charities, but more likely to consist of just one unpaid member of staff, to operate out of their house or associated building on their property, to rely on paper records rather than a computer, and not to carry out post-release monitoring. Collectively, these characteristics are consistent with the image of a passionate hedgehog enthusiast operating from their home address in conjunction with volunteer helpers whilst working full- or part-time.

Furthermore, most recent growth in the size and structure of this rehabilitation community was associated with small hospitals: of the 137 hospitals whose establishment time was known, 94 (68.6%) were <5 years old (77 small, 15 medium-sized, and 2 large). Despite their prevalence within the community, however, small hospitals only accounted for 4.8% of 25,540 hedgehogs admitted in 2016 (n = 174 rehabilitators); in comparison, medium-sized and large hospitals accounted for 12.4% and 82.8% of admissions, respectively. As such, it could be argued that small hospitals make a relatively minor contribution to the rehabilitation of hedgehogs in Britain. Yet, there are possible advantages associated with having a high density of small hedgehog hospitals operating throughout the country, but also limitations.

First, one factor that may increase the likelihood that an animal survives the rehabilitation process is the speed with which it receives care after having been discovered by a member of the public (hereafter ‘finder’). Having a hospital close to the animal’s initial location is likely to reduce transport times, especially if the finder was prepared to take it to the hospital themselves. In these circumstances, affected individuals could potentially be removed ‘out of harm’s way’ more rapidly (e.g., out of direct sunlight, away from potential predators/scavengers, reduced risk of flystrike) and could receive appropriate first aid within a couple of hours of having been discovered, although it must be acknowledged that they may have been injured or unwell for considerably longer. At one extreme, this ‘time to care’ may be reduced even further if the finder was the person who caused the injury in the first place (e.g., with a garden strimmer) or who, for example, disturbed a nest of hoglets while gardening. However, these early advantages are potentially dependent on finders seeking advice from rehabilitators about the appropriate course of action; if these are difficult to contact, as may be the case for those hospitals that do not have a social media presence, then the affected animal may simply be left in an adverse location for some further period of time. In addition, there is anecdotal evidence that members of the public can become frustrated when rehabilitators ask them to deliver the animal to their hospital for treatment, especially if the distance required is large. At one extreme, this could potentially mean that the animal is subsequently ignored and never receives any treatment, or that the finder posts a negative comment on social media which could tarnish the reputation of the rehabilitator/hospital.

It is important to note that the number of hedgehog rehabilitators identified in this study is substantially lower than the >1700 veterinary practices in Britain [58]. However, there is often a degree of confusion relating to the obligation veterinary surgeons are under when it comes to treating wild animals in this country. The Royal College of Veterinary Surgeons code of professional conduct states that “all veterinary surgeons in practice must take steps to provide 24-hour emergency first aid and pain relief to animals according to their skills and the specific situation” (www.rcvs.org.uk/setting-standards/advice-and-guidance/code-of-professional-conduct-for-veterinary-surgeons/supporting-guidance/24-hour-emergency-first-aid-and-pain-relief/, accessed on 10 November 2022). This treatment may be administered free of charge, but they may also be able to claim payment for this emergency initial treatment through a memorandum of understanding between the RSPCA and the British Veterinary Association. After this initial 24-hour period, some practitioners may then expect to be paid for further treatment, whereas others may be working in partnership with local hedgehog rehabilitators, such that they may offer further care at a reduced price or for free. This collaboration enables hedgehogs to be cared for outside of veterinary practices where they would be surrounded by domestic animals and humans [58]. However, this distinction between free treatment in the initial 24-h emergency period, but not necessarily afterwards, has led to a degree of confusion by members of the public, including conflicting information appearing on social media that vets are obliged to treat hedgehogs and other wild animals for free regardless of the timescale involved. Consequently, this has led to a degree of antipathy towards vets from some quarters. In addition, there is also a perception that vets, and some rehabilitation organisations as well, are “too quick to euthanise.” As a result, some rehabilitators recommend that members of the public should take injured hedgehogs to them in the first instance rather than to a vet/other rehabilitation organisation, as the latter are considered to be likely to euthanise an individual even if it presents with survivable injuries. Paradoxically, this would increase the amount of time that an animal requiring veterinary attention would have to suffer before actually being examined by a vet. 

The second potential advantage associated with decreased distance between finding location and hospitals is that rehabilitated individuals may be more likely to be returned to their original site. However, even this is not always possible, as householders may refuse to have the animal returned to their garden and/or the rehabilitator may deem a site to be unsafe (e.g., due to the presence of badgers or busy roads) such that an alternative location is required. At present, there are insufficient data available on the specific release practices of different hospitals, although it is clear that hedgehogs are commonly released at alternative locations for practical or welfare reasons. Whilst this may not affect post-release survival rates significantly (*sensu* [26]), it may influence patterns of gene flow [63,64] and possibly disease transmission within and between populations [65,66].

Despite these putative advantages, small hospitals are also likely to have their limitations. As outlined above, 56.5% of the small hospitals that replied to the questionnaire survey stated that they did not have any social media accounts, and only 17.6% had a business social media account. Furthermore, 125 of the 304 rehabilitators identified, including 96 of 109 small hospitals, were only identified by snowball sampling. These data imply that members of the public may, therefore, not be able to easily find their nearest rehabilitator if they do not have a website/social media presence. In addition, small hospitals were associated with a range of characteristics that could affect their ability to maximise release rates. For example, only 4.9% of small hospitals had one or two paid staff, only 1.9% had three or more volunteers, 94.4% were located in the rehabilitator’s private residence, and 2.8% stated they had no veterinary care. Furthermore, practitioners within small hospitals also need to carry out a wide range of ancillary activities beyond looking after the animals in their care, such as maintaining an online presence, training volunteers, and raising funds. As such, small hospitals may experience significantly greater challenges with, e.g., housing hedgehogs in hygienic conditions, minimizing stress (*sensu* [67]), managing disease spread, and may also have limited capacity at certain times of the year (e.g., when the number of orphaned juveniles is high or during heatwaves), although they are likely to be in contact with other nearby rehabilitators who may have space.

At present, however, all of these putative benefits and concerns are conjecture, and may be unrealised and/or unwarranted, principally because of a lack of data on hedgehog rehabilitation in Britain (either collectively or from individual rehabilitators) to substantiate or refute these claims. This is, in part, due to the difficulties associated with identifying practitioners, but also obtaining access to data for analysis. The latter is further exacerbated by the fact that most hospitals/centres do not keep fully computerised records, which means that collating information about numbers admitted, underlying reasons for admission, cause-specific release rates, and time in care before death, euthanasia, or release is difficult. Major reasons for this relate to the constraints associated with recording data electronically where money and manpower are limited, meaning that those data which are accessible tend to be from the larger centres, but also because hospitals/centres in Britain are not required to collate such data or make it available to researchers. One potential, but highly contentious, option would be to make wildlife rehabilitation a licensed activity, with the submission of data compulsory. However, such a scheme would require funding, and smaller organisations argue that this additional cost would mean that they would have to cease operating. As such, additional methods for collecting such data within the current structure of the rehabilitation community need to be identified.

### The Number of Admissions Relative to Population Size

Extrapolating from the data derived from the questionnaire survey, we estimate that a total of 40,000–59,000 hedgehogs were admitted to the 304 active wildlife rehabilitators identified in 2016. However, the disparity between these estimates suggests that they are sensitive to the modelling approaches used. For example, Model 2 was particularly affected by applying the proportion of large hospitals identified in the online searches (49.2%) in order to estimate the corresponding number in the sample of 120 hospitals for which no data were available; this implies that we had missed 59.0 large hospitals in these initial searches. We consider this to be unrealistic; however, large rehabilitators typically had business or personal social media accounts (88.9%), meaning that they were relatively straightforward to identify. This is also reflected in the fact that only three large hospitals were identified by snowball sampling. Furthermore, the number of active rehabilitators used in deriving these estimates was substantially lower than the ~600 carers purported to be held on the BHPS’s directory, although this organisation does not have detailed information about their status (i.e., whether they are currently practicing, and if so, how many hedgehogs they take in each year). However, the fact that we were not able to identify such a large number of rehabilitators in this study does potentially suggest that these undetected individuals/organisations may not actually be active, but that, if they are practicing, they are likely to be dealing with small numbers of hedgehogs. For these reasons, we consider the lower estimate of 40,000 is more plausible, but accept that it may be higher.

Regardless, this estimate is substantially greater than those from previous studies. For example, [4,11] reported estimates of 30,000–40,000 and 71,000 admissions per annum across the full range of bird (>200 species [68]) and mammal (>40 species [56]) species in Britain, respectively. It is also substantially larger than the number of West European hedgehogs admitted to wildlife centres in other parts of its range (n = 490 in the 5-year period from 2009–2013 in three rescue centres in eastern Spain [69]; n = 16,967 in the 10-year period from 2010–2019 in 34 rescue centres in the Czech Republic [70]; n = 740 in the 17-year period from 2002–2019 to the two main wildlife hospitals in northern Portugal [71]). Furthermore, our estimate does not explicitly include the 31,000 hedgehogs admitted to veterinary surgeons each year [58], although there is likely some degree of overlap, since individual hedgehogs will be taken by rehabilitators to vets for treatment and/or vets will pass on hedgehogs to rehabilitators for long-term care and for release. It does not, however, include those hedgehogs taken directly to vets which are then euthanised or die naturally in the practice.

The most recent estimate of the pre-breeding hedgehog population in Britain is 879,000 [44,56]. If we assume, based on the data from the RSPCA, which are the only data available at the current time, that approximately 50% of hedgehogs admitted to hospitals survive to be released [4], our results suggest that rehabilitators may collectively be saving 20,000 hedgehogs that would otherwise perish, a number equivalent to 2.3% of the national pre-breeding population. However, the causes for admission given by finders (Bearman-Brown and Baker, unpublished data) do suggest that most hedgehog casualties probably originate from urban areas, and that a large proportion of affected animals are juveniles [72]). If we assume that the pre-breeding population of hedgehogs associated with gardens and other urban green spaces is 200,000 [44], the adult sex ratio is 1:1, each adult female produces one litter a year, and mean litter size is 4.5 [36], this figure would equate to 10.0% of the pre-breeding or 3.1% of the post-breeding urban population. Given that presence–absence data from different habitats within urban areas suggest that this population has declined by approximately 2.0% per annum between 2003–2017 [73,74], it can be argued that hedgehog rehabilitation could have been an important component affecting the rate at which the urban population has been declining in the last two decades, rather than being merely a service related to animal welfare, as some authors have suggested previously [14,17,18,19]. It is important to acknowledge, however, that this estimate has been based on assumptions that require further examination once more detailed data are available from a much broader suite of hospitals/centres.

Furthermore, those data from the RSPCA and for those hospitals where we had a continuous set of data across the five-year survey period suggest that 2016 was a rather atypical year; in the former, the number admitted in 2016 was 22% higher than the average across the 13-year time period, whereas in the latter, the number admitted was significantly higher than in 2012 and 2013. At one level, the number of admissions would be expected to vary inter-annually, but, for a population that is assumed to be declining [44], we might expect to observe a general downward trajectory (assuming that the capacity of these hospitals has been approximately constant over time). In contrast, the pattern observed has been one for a general increase in admissions (Figure 4d and Figure 5), with periodic peaks (Figure 5). The underlying reasons for these inter-annual fluctuations are unclear, but could be related to broad-scale changes associated with over-winter hibernation mortality, reproductive output, invertebrate prey availability, public awareness, risks related to anthropogenic activities, and/or inclement weather. As such, future research needs to focus on identifying how such ecological and human-mediated factors affect population dynamics and patterns of admission into wildlife hospitals.

## 5. Conclusions

This study was the first to attempt to directly estimate the number and characteristics of practitioners rehabilitating hedgehogs in Britain. A minimum of 304 rehabilitators were identified: most (n = 109) hospitals admitted less ≤50 hedgehogs annually, whereas most hedgehogs (82.8%) were admitted to the smaller number of large establishments (n = 36). However, the growth of the hedgehog rehabilitation community was mostly associated with the creation of these smaller hospitals. Overall, the collective number of hedgehogs admitted to (40,000–59,000) and potentially released from (20,000–29,500) these wildlife hospitals was large relative to the size of the pre-breeding population and the estimated annual rate of decline, especially if we assume that most individuals originated from urban areas. This implies that wildlife rehabilitation has potentially been an important factor in the dynamics of hedgehog populations in the last two decades. However, further information is still required regarding many facets of the rehabilitation process, as well as establishing how extrinsic ecological and anthropogenic factors affect the numbers of hedgehogs admitted to wildlife hospitals. Such future work would benefit particularly from facilitating means by which rehabilitators could store standardised information electronically.

## Figures and Tables

**Figure 1 animals-12-03139-f001:**
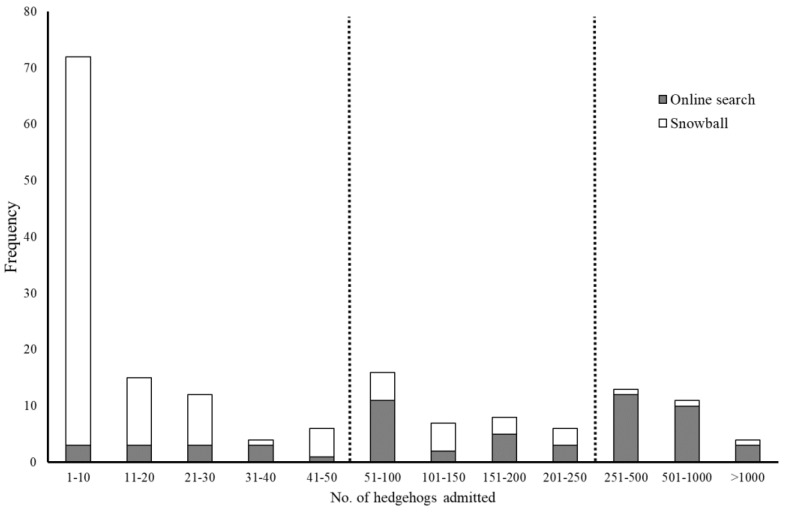
Frequency distribution of small, medium-sized, and large wildlife hospitals/rehabilitation organisations identified from original online searches (n = 59) versus snowball sampling (n = 115). Vertical lines indicate size categories defined retrospectively on the basis of the number of hedgehogs admitted in 2016: small ≤ 50 admissions; medium-sized = 51–250 admissions; and large ≥250 admissions.

**Figure 2 animals-12-03139-f002:**
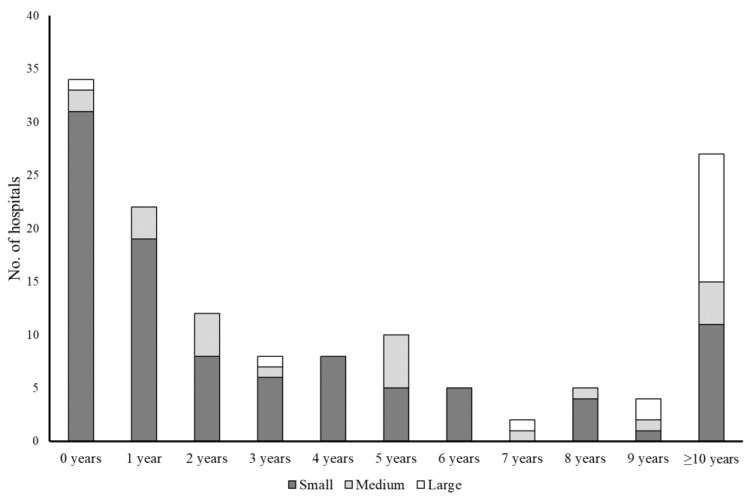
Number of years that small (≤50 admissions; n = 98), medium-sized (51–250 admissions; n = 22), and large (>250 admissions; n = 17) hospitals had been established in 2016.

**Figure 3 animals-12-03139-f003:**
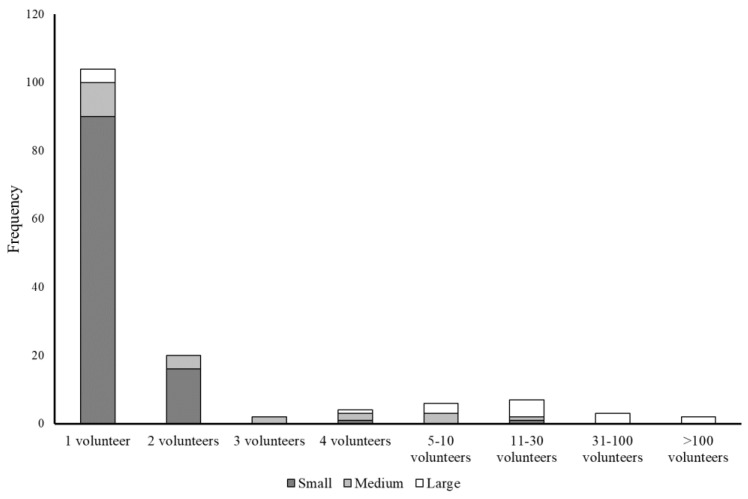
Frequency plot of the number of unpaid staff (volunteers) working at small (≤50 admissions; n = 108), medium-sized (51–250 admissions; n = 22), and large (>250 admissions; n = 17) hospitals in 2016.

**Figure 4 animals-12-03139-f004:**
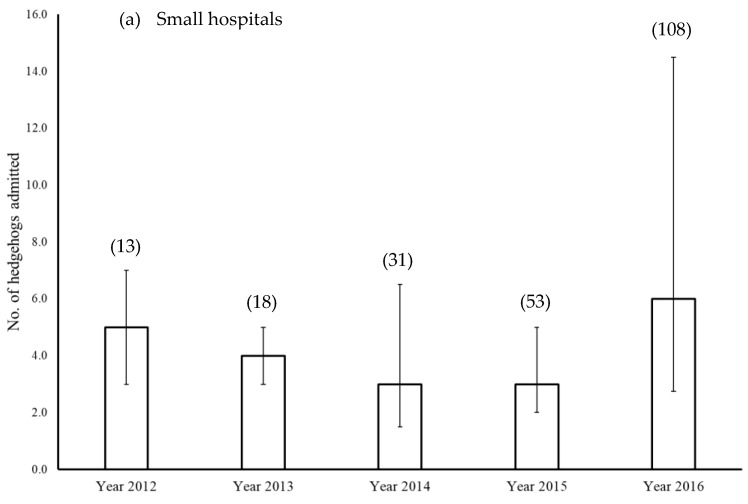
Median (±IQR) number of hedgehogs admitted annually to (**a**) small (≤50 admissions), (**b**) medium (51–250 admissions), and (**c**) large (>250 admissions) hospitals each year in the five-year period from 2012–2016, inclusive; figures above columns indicate sample sizes. (**d**) Number of hedgehogs admitted annually for those hospitals (n = 28) that provided data for all five years; letters above columns indicate post hoc groups from a Friedman test.

**Figure 5 animals-12-03139-f005:**
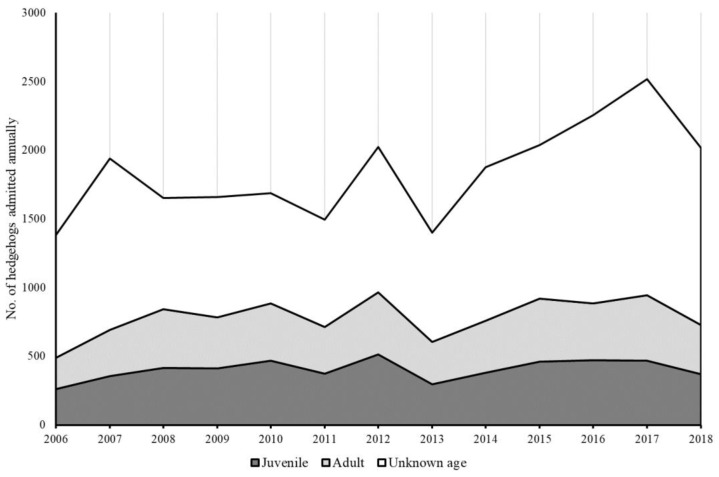
Number of juveniles, adults, and hedgehogs of unknown age admitted annually across the four wildlife hospitals in England run by the Royal Society for the Prevention of Cruelty to Animals during the period from 2006–2018, inclusive.

**Table 1 animals-12-03139-t001:** Summary of the median (±IQR) number of hedgehogs admitted to small (≤50 admissions per annum), medium-sized (51–250 admissions), and large (>250 admissions) hospitals in 2016 based on whether they had been identified in the original online search (n = 59) versus those identified by snowball sampling (n = 115). Figures are also presented for all hospitals combined.

	Small	Medium	Large
Online	25.0 (10.0–32.0) (n = 13)	97.0 (67.5–156.5) (n = 17)	500.0 (346.0–701.0) (n = 29)
Snowball	6.0 (2.0–12.8) (n = 96)	121.0 (59.0–143.0) (n = 12)	235.0 (201.0–582.0) (n = 7)
Combined	6.0 (2.5–16.0) (n = 109)	97.0 (63.5–145.0) (n = 29)	478.0 (261.0–645.0) (n = 36)

**Table 2 animals-12-03139-t002:** Estimated number of hedgehogs admitted to all wildlife hospitals active in 2016 (n = 304); extrapolations are based upon data supplied by 59 of 159 hospitals identified from an online internet search and 115 of 125 hospitals identified from subsequent snowball searching. Hospital size was categorised on the basis of the number of hedgehogs admitted in 2016: small ≤50 admissions; medium = 51–250 admissions; and large ≥250 admissions. Model 1 assumed that there were no significant differences in (i) the relative distribution of hospitals of different sizes nor (ii) the median number of hedgehogs admitted annually between hospitals identified in the original online search versus snowball sampling; estimates were, therefore, based on the combined data. Model 2 assumed that there were significant differences in both of these components between hospitals identified in the original online search versus snowball sampling; estimates were, therefore, based on each subset of data separately.

**Model 1: Online Internet Search and Snowball Sampling Data Combined**			
	**Small**	**Medium**	**Large**
Total number of hospitals identified (N_O_ = hospitals identified by online search; N_S_ = hospitals identified by snowball search)	304 (N_0_ = 179, N_S_ = 125)		
Total number of hospitals where number of hedgehogs admitted was known	174 (N_0_ = 59, N_S_ = 115)		
Number (%) of hospitals in each size category where the number of hedgehogs admitted was known	109 (62.6%)	29 (16.7%)	36 (20.7%)
Absolute number of hedgehogs known to have been admitted in 2016 within each hospital size category	1226	3169	21,145
Number of hospitals where number of hedgehogs admitted in 2016 was not known	130 (N_0_ = 120, NS = 10)		
Estimated number of hospitals in each size division where data on the number of hedgehogs admitted was unknown	81.4 (130 × 0.626)	21.7 (130 × 0.167)	26.9 (130 × 0.207)
Median number of hedgehogs admitted per hospital within each hospital size category (see Table 1)	6	97	478
Estimated number of additional hedgehogs admitted in 2016 within each hospital size category	488 (81.4 × 6)	2105 (21.7 × 97)	12,858 (26.9 × 478)
Estimated total number of hedgehogs admitted in 2016 within each hospital size category	1714	5274	34,003
Estimated total number of hedgehogs admitted in 2016	40,991		
**Model 2: Online Internet Search and Snowball Sampling Data Treated Independently**			
	**Small**	**Medium**	**Large**
**(a) Hospitals identified through online search**			
Total number of hospitals identified	179		
Total number of hospitals where number of hedgehogs admitted was known	59		
Number (%) of hospitals where number of hedgehogs admitted was known	13 (22.0%)	17 (28.8%)	29 (49.2%)
Known number of hedgehogs admitted within each hospital size category	299	1833	18,070
Median number of hedgehogs admitted per hospital within each hospital size category (see Table 1)	25	97	500
Number of hospitals where number of hedgehogs admitted was not known	120		
Estimated number of hospitals in each size division where data on the number of hedgehogs admitted was unknown	26.4 (120 × 0.220)	34.6 (120 × 0.288)	59.0 (120 × 0.492)
Estimated number of additional hedgehogs admitted in 2016 within each hospital size category	660 (26.4 × 25)	3356 (34.6 × 97)	29,500 (59.0 × 500)
Estimated total number of hedgehogs admitted in 2016 within each hospital size category	959	5189	47,570
Estimated total number of hedgehogs admitted in 2016	53,718		
**(b) Hospitals identified through snowball sampling**			
Total number of hospitals identified	125		
Total number of hospitals where number of hedgehogs admitted was known	115		
Number (%) of hospitals where number of hedgehogs admitted was known	96 (83.5%)	12 (10.4%)	7 (6.1%)
Known number of hedgehogs admitted within each hospital size category	927	1336	3075
Median number of hedgehogs admitted per hospital within each hospital size category (see Table 1)	6	121	135
Number of hospitals where number of hedgehogs admitted was not known	10		
Estimated number of hospitals in each size division where data on the number of hedgehogs admitted was unknown	8.4 (10 × 0.835)	1.0 (10 × 0.104)	0.6 (10 × 0.061)
Estimated number of additional hedgehogs admitted in 2016 within each hospital size category	50 (8.4 × 6)	121 (1.0 × 121)	81 (0.6 × 135)
Estimated total number of hedgehogs admitted in 2016 within each hospital size category	977	1457	3156
Estimated total number of hedgehogs admitted in 2016	5590		
**(c) All hospitals combined**			
Estimated total number of hedgehogs admitted in 2016 within each hospital size category	1936	6646	50,726
Estimated total number of hedgehogs admitted in 2016	59,308		

**Table 3 animals-12-03139-t003:** Summary of the characteristics of small (≤50 hedgehogs admitted in 2016), medium-sized (51–250 hedgehogs admitted), and large (>250 hedgehogs admitted) wildlife hospitals (n = 148). Sample sizes vary between individual analyses if respondents had not answered that question.

Characteristics		Small (n = 108)	Medium (n = 22)	Large (n = 18)	Chi-Squared Results
Registered charity (n = 148)	No	96.3%	86.4%	27.8%	X^2^_2_ = 61.98, *p* < 0.001
	Yes	3.7%	13.6%	72.2%	
No. of paid staff (n = 146)	% of hospitals with paid staff	4.7%	0.0%	44.4%	-
	Mean no. of paid staff (range)	1.2 (1–2) ^1^	-	6.4 (1–30)	
No. of unpaid staff (n = 148)	1 volunteer	83.3%	45.5%	22.2%	X^2^_4_ = 74.70, *p* < 0.001
	2 volunteers	14.8%	18.2%	-	
	3 or more volunteers	1.9%	36.4%	77.8%	
Veterinary care (n = 148)	None	2.8%	-	-	X^2^_8_ = 13.03, *p* = 0.111
	Work with external vet practice	81.5%	100.0%	88.9%	
	Work with external rescue/hospital	12.0%	-	-	
	I am a veterinary nurse/vet	1.9%	-	-	
	Have an onsite vet	1.9%	-	11.1%	
Length of time established (n = 137)	≤5 years	78.6%	68.2%	11.8%	X^2^_2_ = 30.03, *p* < 0.001
	>5 years	21.4%	31.8%	88.2%	
Housing (n = 148)	Building in private grounds	3.7%	18.2%	11.1%	X^2^_4_ = 41.53, *p* < 0.001
	Personal residence	94.4%	77.3%	50.0%	
	Purpose-built facility	1.9%	4.5%	38.9%	
Social Media (n = 148)	No social media account(s)	56.5%	4.5%	11.1%	X^2^_4_ = 52.46, *p* < 0.001
	Only personal account(s)	25.9%	18.2%	5.6%	
	Business and/or personal account(s)	17.6%	77.3%	83.3%	
Record-keeping (n = 132)	Paper (partly or fully)	78.3%	45.5%	50.0%	X^2^_2_ = 12.42, *p* = 0.002
	Computer (partly or fully)	21.7%	55.5%	50.0%	
Post-release monitoring (n = 145)	No	58.9%	38.1%	23.5%	X^2^_2_ = 9.17, *p* = 0.010
	Yes	41.1%	61.9%	76.5%	

^1^ Two small hospitals based at higher education establishments were excluded from these figures as they listed the number of paid staff as “lots” and “4000” which presumably refers to the students at these establishment.

## Data Availability

The data presented in this study are available on request from the corresponding author.

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
