# Peer review of "An Estimate of the Scale and Composition of the Hedgehog (Erinaceus europeaus) Rehabilitation Community in Britain and the Channel Islands"

_animals, 2022, doi:10.3390/ani12223139_

Round 1

Reviewer 1 Report

This paper is an original and useful contribution to our understanding of the scale and potential population influence of wildlife rehabilitation, as well as providing insights into the workings of the rehabilitation centres. The paper is very well-written and presented, with appropriate and clearly explained analyses.

The authors should consider the following minor points: 

Lines  14 and 29:  148 responded to the questionnaire, but this is the number that supplied information about both their structure and the number of hedgehogs they admitted in 2016 (line 321). When estimating total hedgehog admissions in 2016 there was a sample size of 174 (line 291). So you must have had a response. Perhaps this should be rephrased? e.g. 'Of 174 responses, 148 gave information on both their structure and intake of hedgehogs.' 

Line 88 'gild' should be spelled 'guild'.

Lines 89-92 'Most recently, the British hedgehog population was estimated to number approximately 0.88 million individuals [44,56], down from 1.56m in the mid-1990s [57], triggering its status to be upgraded to Vulnerable [56]. Despite this substantial decline, . . .' 

These two population estimates were both quite unreliable and one might argue that one should only allow them as evidence of a substantial decline when also taking into account the various downward trends detailed in Wembridge et al (2022) (ref. 44). This may be pedantic and I understand the need to be concise, but I think that you should be clearer that the evidence for decline is not simply these two estimates.

Line 516 '... that hedgehog rehabilitation has been an important ...' should be "...that hedgehog rehabilitation could have been an important..'

Line 539 I think 'smaller' would read better than 'small'

Should the questionnaire be made available as a supplementary item?

Finally, throughout the reference list specific names should not start with a capital letter.

Reviewer 2 Report

This is a very interesting paper and one that has very useful potential application in the field of wildlife rehabilitation. The study is quite novel and has managed to include an amazing amount of probably unique data from wildlife rescue centres, including the smaller ones. It is generally well written. I do perhaps have some concerns about the conclusions being drawn being a bit overly positive and the impact this could inadvertently have on hedgehog rehabilitation, these concerns are addressed below. 

Line 26 and 27 and 32 – I might change ‘wildlife hospital’ to ‘centre’ in the summary and abstract to avoid confusion. Hospital perhaps implies much more than is provided by most of these places (see comments on line 151)

Line 36 (and 17) – I might change ‘would’ to ‘could’ relating to the quoted 50% release rate. It’s an RSPCA figure and may not be achieved elsewhere.

Line 61 – what ‘impacts’ do you mean? Animal welfare, population loss?

Line 69 – can you add ‘….successfully rehabilitated….’ in here? 

Lines 73-76 – Can there be some mention of variations in experience, training and standards in here, especially in countries such as Britain where there is no licencing and the only regulation is existing animal welfare legislation that is rarely applied.

Line 91 – ‘Vulnerable’ on what? Mammal Society’s Red List for British mammals? Just need to make this distinct from the IUCN list in which hedgehogs are really not a concern.

Line 98 – I might lift the euthanasia from here and have a separate sentence dealing specifically with that. Hedgehogs are protected so can really only be killed to prevent suffering (‘what would be described as ‘an act of mercy’ for other species, for example the Protection of Badgers Act’). As most hedgehog euthanasia requires veterinary involvement (direct or via prescribed drugs), this would sit better after the veterinary bit on line 100. Something like ‘As part of this care, hedgehogs may be humanely killed (euthanased) to prevent suffering and/or where they are unsuitable cases for release back to the wild.’

Line 103/104 – I’d stress that the legality and ethics of long-term captivity of a wild hedgehog are highly debated (ref 59 and maybe other references around the ethics of this)

Line 107 (109, 113, 118) – really not convinced the web addresses are needed or are useful here for the big organisations (unsure what ‘Animals’ house style is, but it seems an unusual thing to include). If they are included, they need to be the same font and format.

Line 115 – where is that >1000 figure from?

Line 127 – the Vale courses should be a reference rather than a web address in the text. 

I wonder here again if there should be some mention of the variations in standards of care across these centres and the lack of reliable information around outcomes. Risk of zoonoses, risk of transmission of infections within centres, lack or availability of veterinary care, are all important.

Line 151 – I might add something like: “the term ‘hospital’ does not necessarily confer any sort of medical standard to that premises”. 

Lines 153/154 – the web addresses do work Ok here

Line 165 – maybe ‘chain-referral sampling’ in brackets after snowball sampling, just to make that term clear for all readers

Line 172 – SmartSurvey needs a trademark sign

Line 176 – you’ve already used BHPS in line 118 so not needed in full here

Line 193 – is it Ok to call this ‘post release monitoring’? You’ve only really asked about marking, there’s no detail as to how this marking was used. Maybe rephrase in some way, perhaps ‘….. undertook marking for post-release monitoring ……’

Line 219 onwards – is this bit (from ‘as there was evidence…..’) really needed here in M&Ms?

Line 276 – suggesting you were missing data from 130 establishments only is maybe not correct. You missed data from 130 of your original identified establishments, but as you found lots more establishments via snowballing, you might also have missed some more and should acknowledge that. I’d maybe change this to something like: “Data on the number of admissions in the benchmark year for 174 of 304 hospitals (57.2%: n = 59 + 115) was collected. Data was missing from a further 130 known establishments (n = 120 + 10).” I’d avoid using ‘we’.

Line 335 (and Table 3) – again this is ‘marking for post-release monitoring’ not really evidence of actual monitoring

Lines 402-404 – I think your comments here are a very positive reflection. The large number of small rehabilitators identified only via snowballing and working by their own admission in less-than-ideal conditions, with limited professional contacts and some with no veterinary input could equally be taken as an indication of hedgehogs being brought into circumstances where they were not likely to be adequately cared for. There’s quite a bit of info in Table 3 including your statistical analyses that says a lot about the differences between centres of different sizes, perhaps you could attempt to interpret what these differences might mean for say quality of record keeping, veterinary care, engagement with the broader ‘rehabilitation’ profession, marking pre-release, and how all this might impact on animal welfare, release rates and survival?

Paragraph line 415-429 – I’m not sure I agree with all this. I really don’t think ‘life-saving care’ for adult hedgehogs is very often provided by rehabilitators, especially the smaller ones. This sort of care would be proper first aid (fluids, analgesia, heat, maybe oxygen) which could only be provide in a vet practice or in a centre with good experience and training working closely with a vet. Most injured adult hedgehogs would I hope receive some direct veterinary care, with rehabilitators being important for ongoing care.  The availability of rehabbers for perhaps that initial support of finders is important, as it also is for that ongoing rehabilitation and release. The better centres without their own vet would probably direct the finder to the nearest vet rather than asking them to bring the animal to the rehab centre. For neonates it’s probably a bit different as veterinary/medical care isn’t usually required, and good rehabilitators can indeed offer life-saving supportive care. Not sure how you reword all this, but be careful it doesn’t sound like going into all these places is necessarily resulting in best standard care (I appreciate this same criticism can be directed at vets too).

Lines 434-439 – this bit about veterinary obligations is confused and could be better referenced. Hopefully this will help:

Vets are obliged to provide first aid and emergency care to all animals independent of any agreement with BVA/RSPCA, see RCVS guidance on first aid and emergency care: https://www.rcvs.org.uk/setting-standards/advice-and-guidance/code-of-professional-conduct-for-veterinary-surgeons/supporting-guidance/24-hour-emergency-first-aid-and-pain-relief/

The BVA/RSPCA is memorandum of agreement around payment only (not the RCVS need to treat) and in theory only applies to BVA members. Vets provide care under this free of charge for animals <1kg presented during normal hours and may obtain an RSPCA log number for some funding if >1kg or out of normal hours. Guidance at : https://www.bva.co.uk/resources-support/practice-management/rspca-payments-for-initial-emergency-treatment-iet-guide/ (and attached)

All vets are legally obliged to provide primary care (it’s not about them being ‘willing’) but are under no obligation to provide more than that. Anything beyond primary care should normally be expected to be paid for, though some vets do provide additional care for free. 

Vets have traditionally been considered to lack knowledge relating to British wildlife and be disinterested in supporting rehabilitators. This has led to the sort of apathy/conflict you suggest. In the last decade however, this has really changed with lots of veterinary information (e.g. BSAVA wildlife manual including 2nd 2016 edition, Born Free Foundation vet resources) and many vets with post graduate qualifications in wildlife and zoo animals. With the smaller rehabbers one of the biggest issues, as seen on social media, is the need to euthanase many of these animals and their fear that a vet will rightly do that. The conflict between vets and rehabbers here is actually just the same as with the pro euthanasia larger ‘professional’ centres and the smaller home-based centres and this is very much seen in comments on social media (e.g. ‘Wildlife Rehabilitation Professionals’ Facebook site). 

Line 441- I’m not sure ‘perceived’ is valid (unless it’s a direct quote from Emily’s paper), veterinary practices are absolutely not suitable places to keep wild animals for anything other than initial primary care (noise, risk of infection transfer in all directions, unlikely to have adequate facilities to be safe for animals or staff). 

Line 453 – I think ‘not always possible’ should be ‘considered/perceived’ as not always possible. Good practice would be to always put adult animals back where they were found and soft release juveniles at appropriate sites.

Line 459 – spread of disease as well as gene flow

Paragraph line 460-464 – whilst I accept that we don’t have the evidence to compare small and medium/large centres in terms of outcome, I do think it’s amiss not to mention that small, reasonably new centres, are likely to lack knowledge, facilities and training and that some of them having no veterinary involvement is worrying. Also risk of disease spread between and to hedgehogs in less-than-ideal housing. Just a mention of animal welfare would be good! I might also go back to the fact that the majority of animals are passing through the bigger centres, so although the small centres are numerous, they are less important overall than they might appear. 

Line 504 – the 50% figure is assuming that the care received is as good as the RSPCA, I think that’s a big presumption.

Paragraph lines 503-519 – again I find this interpretation perhaps a bit ‘rose tinted’. We don’t know if many of these hedgehogs needed to be admitted to rehab centres at all (somewhere perhaps you need to acknowledge that hedgehogs are easy to spot, pick up and keep, and that this may be to their detriment compared to other British wild animals),  we don’t know what the standard of care is, we don’t know how many are released, or how many subsequently survive, or how those survival rates compare to those of wild hedgehogs of a similar age. Could this paragraph be reworked a bit with positives and negatives? I just worry that otherwise this paper sounds like ‘rehabilitation’ is a great thing in all circumstances and that could potentially be dangerous. You could acknowledge that more questions about knowledge, standard of care etc. could be useful in any follow up study. Your last paragraph and conclusion are then a good follow on from this.
